# Analysis of the Difference in College Students’ Experience of Family Harmony before and after the COVID-19 Outbreak

**DOI:** 10.3390/ijerph19106265

**Published:** 2022-05-21

**Authors:** Qisheng Zhan, Shuya Zhang, Qin Wang, Lisha Zhang, Zhengkui Liu

**Affiliations:** 1School of Education, Tianjin University, Tianjin 300350, China; zsy_6@tju.edu.cn (S.Z.); wangqin123@tju.edu.cn (Q.W.); lishazhang@tju.edu.cn (L.Z.); 2Institute of Psychology, Tianjin University, Tianjin 300350, China; 3Institute of Psychology, Academy of Science, Beijing 100020, China; liuzk@psych.ac.cn

**Keywords:** China, college students, COVID-19 outbreak, experience of family harmony

## Abstract

Background: China implemented a home quarantine policy in the early days of the COVID-19 pandemic. At the same time, college students stayed at home for a long time, facing their parents and being directly exposed to family affairs every day. Thus, the effects of the COVID-19 pandemic and home quarantine on college students’ experience of family harmony are worth discussing. Objectives: In this study, we aimed to explore whether there was any difference in college students’ experience of family harmony before and after the COVID-19 outbreak. Methods: Participants in this study were undergraduates from a university in Tianjin. They completed the college students’ experience of family harmony questionnaire (CSEFHQ) before and after the COVID-19 outbreak (December 2019 and March 2020). A total of 215 participants (96 men and 119 women) completed the whole test. Results: The paired sample *t*-tests showed that the scores on seven dimensions of CSEFHQ: getting along (t = 5.116, *p* < 0.001), conflict (t = 6.442, *p* < 0.001), sharing (t = 5.414, *p* < 0.001), self-isolation (t = 3.014, *p* < 0.01), help-seeking (t = 5.353, *p* < 0.001), avoidance (t = 6.010, *p* < 0.001), support-providing (t = 5.818, *p* < 0.001), and the total scores of CSEFHQ (t = 6.496, *p* < 0.001) were all significantly reduced after the COVID-19 outbreak, while the scores on the other two dimensions, undertaking housework (t = 1.379) and indifference (t = 1.765), did not change significantly. Conclusions: The college students’ experience of family harmony was significantly worse after the COVID-19 outbreak. These results can be used to improve the level of family harmony of college students during the pandemic and improve their quality of life.

## 1. Introduction

The COVID-19 pandemic broke out in December 2019 and quickly swept the world, with the virus affecting the global economy and everyone’s daily lives. Research early in the pandemic showed that more than half of participants thought the psychological impact of the outbreak was moderate to severe, and about one-third of participants reported moderate to severe anxiety [1]. To curb the spread of COVID-19 and address the public health problems of citizens, China implemented a home quarantine policy in the early days of the COVID-19 pandemic. Physical distancing and community containment measures were implemented nationwide, which required people to reduce the use of public transportation and public gatherings and study and work from home when possible [2]. The containment policies that restricted mobility, as well as the pandemic itself, affected people’s mental health [3,4]. With major Chinese cities and schools closing at all levels indefinitely, uncertainties and potentially negative factors adversely affected students’ mental health [1]. Studies have shown that college students generally showed higher levels of anxiety and depression [1], generalized anxiety disorders, depressive symptoms, and sleep quality problems during the pandemic [5]. The COVID-19 containment policy has produced different psychological outcomes in different populations [6].

In addition to affecting college students themselves, the pandemic also indirectly affected their interpersonal interactions [7]. Home quarantine and the lack of normal social interaction brought a sense of social isolation, while increased negative interactions with family members affected family harmony [7]. Family harmony is an ideal state of family relations which refers to harmonious coexistence in family life [8]. It can be measured by the college students’ experience of family harmony questionnaire (CSEFHQ) [9]. According to the survey, 84.7% of respondents initially reported staying at home for more than 20 h [1], and people spent significantly more time together with their families. In a closed and isolated environment, the conflicts and contradictions between college students and their families may intensify, which may have led to their poor experience of family harmony. Therefore, conducting this study can effectively illustrate the impact of the COVID-19 outbreak on the experience of family harmony among college students.

Asian countries influenced by Confucianism attach great importance to family harmony [8], especially in China. Harmony is an idea that is highly valued and praised in traditional Chinese culture. The cultural inheritance for thousands of years has made the concept of harmony deeply rooted in the hearts of Chinese people. A related study showed that college students with harmonious relationships in their families had a higher degree of trust than those who had quarrelsome relationships in their families [10]. College students with strong family cohesion and fewer conflicts showed better academic, social, and emotional adaptations during college [11]. The individual perception and evaluation of family harmony are different from the relatively objective comprehensive evaluation of family harmony. Family harmony experience is the subjective evaluation of whether the family relationship is harmonious [9]. The experience and perception of family harmony is more important for the development of college students’ physical and mental health.

There is no defined concept of family harmony in Western culture. However, experts study strong families, well-functioning families, healthy families, good families, cohesive families, and so on [12,13,14,15]. In essence, they are consistent with the connotation of the harmonious family, and all of them have the same goal: to achieve family harmony. However, there are some differences in the specific standards, extension, and other aspects [16]. The questionnaire used in our study has five dimensions: the atmosphere of family, responsibility for housework, time-sharing, seeking help, and supporting family members, which is fit for investigating the college students’ experience of family harmony in China.

In this study, we aimed to identify the effects of the COVID-19 pandemic and the home quarantine policy on college students’ experience of family harmony. There are few longitudinal studies comparing changes in individual mental health levels before and after the COVID-19 outbreak [7,17]. The measured time of this study coincided with the two time points before and after the COVID-19 outbreak, and the results of the two tests objectively constituted the basis of the comparative data before and after the COVID-19 outbreak. The research content partly filled the gap in the longitudinal studies of COVID-19. The results reflected changes in family interactions and could partially explain the changes in mental health levels during the outbreak. Experts predict that the COVID-19 pandemic will come to an end by November 2023 [18]. However, it is still currently ongoing, and isolation will continue for a long time. In this study, we compared the differences in college students’ experience of family harmony before and after the COVID-19 outbreak. The results expand the study of family interaction and described specific changes in family harmony. Meanwhile, the decline in mental health levels during the COVID-19 pandemic can be partly explained by our findings.

## 2. Materials and Methods

### 2.1. Survey Tools

This survey has two parts. The first part includes basic information, such as name, gender, age, and grade. The second part is the college students’ experience of family harmony questionnaire (CSEFHQ), which was specially designed for the experience of family harmony of Chinese college students [9]. The structure of the questionnaire is reasonable, and its reliability and validity meet the requirements of psychometry. Therefore, it can be used as an effective tool to evaluate the family harmonious experience of Chinese college students. The Cronbach’s alpha (α) coefficient of this questionnaire was 0.97. The split-half reliability was 0.92, and the test-retest reliability was 0.75 for the total questionnaire. The questionnaire used in this study included 5 modules: the family atmosphere (getting along and conflict), the responsibility of housework (undertaking housework), time-sharing (sharing and self-isolation), seeking help (help-seeking and avoidance), and supporting family members (support-providing and indifference), and 9 dimensions (getting along, conflict, undertaking housework, sharing, self-isolation, help-seeking, avoidance, support-providing, and indifference) which included a total of 56 items. Reverse scoring was used for the four dimensions including conflict, self-isolation, avoidance, and indifference. The scale is shown in Table 1.

### 2.2. Procedures and Participants

The above questionnaire was used to test 225 participants twice, and the time interval between the two tests was 3 months (precisely before and after the COVID-19 outbreak, December 2019 and March 2020). In this study, all participants enrolled in the survey were students who experienced home quarantine. The participants were undergraduates from a university in Tianjin, and 225 participants completed the surveys independently. There were 215 pairs of validated questionnaires after excluding repeated answers and invalid questionnaires, with an effective efficiency of 95.6%.

### 2.3. Ethics

This study received ethical approval from the related Ethics Committee of Tianjin University. Informed consent was obtained from all individual participants included in the study.

### 2.4. Analytical Software

Paired sample *t*-tests were used to compare significant differences in pre-test data and post-test data with SPSS 23.0 statistical software. Before the *t*-tests, all the scores of the reversal scoring items were converted correctly according to the scoring rules.

## 3. Results

### 3.1. Participant Characteristics 

Out of 225 participants, 215 participants had valid questionnaires with an effective efficiency of 95.6%. Participants were 17–22 years old, including 96 men (44.7%) and 119 women (55.3%). See Table 2 for details.

### 3.2. Analysis of the Difference in the Total Scores of CSEFHQ before and after the COVID-19 Outbreak

A paired sample *t*-test was performed on the total scores of CSEFHQ, and the results are shown in Table 3.

As can be seen from Table 3, the mean total score of college students’ experience of family harmony during the pandemic period was less than that before the pandemic, and the *t*-test of paired samples showed significant differences in the measured data before and after the pandemic. It showed that the college students’ experience of family harmony during the pandemic was significantly worse than that before the pandemic.

### 3.3. Analysis of the Difference in 9 Dimensions of CSEFHQ before and after the COVID-19 Outbreak

Paired sample *t*-tests were performed on the scores of 9 dimensions of CSEFHQ, and the results are shown in Table 4.

According to Table 4, all t-values were more than 0.000, indicating that the post-test scores of 9 dimensions (getting along, conflict, undertaking housework, sharing, self-isolation, help-seeking, avoidance, support-providing, and indifference) were all less than the pre-test scores. Among them, the scores of 7 dimensions (getting along, conflict, sharing, self-isolation, help-seeking, avoidance, and support-providing) were significantly lower than those of the post-test. It showed that excluding the dimensions of undertaking housework and indifference, the scores of the remaining dimensions significantly decreased.

### 3.4. Analysis of the Difference in Scores of 56 Items of CSEFHQ before and after the COVID-19 Outbreak

Paired sample *t*-tests were performed on the scores of 56 items, and the results are shown in Table 5.

As can be seen from Table 5, the t-values of most items were more than 0.000, and the test results were significant, which indicated that the college students’ experience of family harmony generally decreased for all items. In the module on the family atmosphere, the post-test scores of 9 items in the getting along dimension were significantly lower than those of the pre-test; in the conflict dimension, except for the first item “family members often quarrel with each other (CO1)”, the related differences of the remaining 6 items were significantly lower than those of the pre-test. Although the related difference of “family members often quarrel with each other (CO1)” was not significant, the related difference of “family members have a cold war with each other (CO2)” was significant.

In the responsibility of housework module, only the scores for the 2 items of “we are willing to spend a lot of energy doing things at home (UN6)” and “everyone in the family does his job” were significantly lower than those of the pre-test, and the related differences of the remaining 5 items were not significant, among which the t-value of “we can share the housework together (UN4)” was less than 0.000.

In the time-sharing module, the post-test scores of 8 items in the sharing dimension were significantly lower than those of the pre-test; in the self-isolation dimension, the related differences of “we don’t have enough time to get along and communicate with each other (SE1)” and “there is little time for family members to spend time with each other (SE4)” were not significant, and the t-value of “we don’t have enough time to get along and communicate with each other (SE1)” was less than 0.000, which is consistent with the impact of the pandemic.

In the seeking help module, only the related difference of “I communicate with my family as soon as possible when something happens to me (HE4)” was not significant, and the post-test scores of the remaining 4 items were significantly lower than those of the pre-test in the help-seeking dimension. In the avoidance dimension, only the related difference of “when I come across something that makes me sad, it’s hard to talk to my family (AV5)” was not significant, and the post-test scores of the remaining 5 items were significantly lower than those of the pre-test.

In the supporting family members module, the post-test scores of 6 items in the support-providing dimension were significantly lower than those of the pre-test; in the indifference dimension, the related differences of “my family and I don’t care about each other (IN1)” and “the family members only care about themselves and ignore the family (IN2)” were not significant. The post-test scores of “I’m self-centered and I don’t care about my family (IN3)” were significantly lower than those of the pre-test, and the t-value of “the family members only care about themselves and ignore the family (IN2)” was less than 0.000.

## 4. Discussion

The present study examined whether there were differences in Chinese college students’ experience of family harmony before and after the COVID-19 outbreak. Through the paired sample *t*-test, the study found that Chinese college students’ general experience of family harmony significantly decreased after the outbreak of the pandemic, especially in the dimensions of getting along, conflict, sharing, self-isolation, help-seeking, avoidance, and support-providing, which are also the poor aspects of family interaction.

In terms of feelings about the family atmosphere, college students were more stressed at home after the COVID-19 outbreak. They thought that the family atmosphere was more depressed and they didn’t want to stay at home, indicating that college students’ subjective feelings about the family atmosphere were significantly worse. In terms of family interaction, all positive interactions such as free expression, love, and humility were reduced, and negative interactions such as cold war, blame, and criticism increased. Although mutual quarrels didn’t change significantly, cold wars increased. This phenomenon fit with the ideal family value of avoiding quarrels in Chinese Confucian [19]. This showed that the impact of the COVID-19 outbreak did not promote a harmonious family atmosphere, but brought more conflicts. The parent–child relationship is the main relationship of the family, which is an important factor affecting teenagers’ life satisfaction and also an important factor to form family harmony [20]. For college students, the time when they would go back to school was uncertain. Staying in a fixed and closed environment for a long time and only facing their parents, college students would inevitably have conflicts with their parents which affect family harmony.

Due to the COVID-19 pandemic, people couldn’t return to work in most places, so they had to work and study online. People spent a long time at home [21], so family members could discuss the division of labor and share housework. However, it is challenging to undertake housework [21]. College students’ willingness to spend a lot of energy doing things at home declined, and they didn’t think that everyone in the family did their job. The division of family members’ labor and the shared family obligations are some of the elements of family happiness and harmony [20]. Housework is emotional work, not simple labor [14]. During the pandemic period, family members did more housework, but their willingness to do housework decreased, indicating that the pandemic harmed the sense of family harmony.

Family interaction that follows the shared model contributes to family harmony [22], and sharing family time is the core of family harmony and happiness. Living at home made college students spend more time with their families, and college students had enough time to accompany them. However, their willingness and behavior to share happiness with their families decreased, and their satisfaction with sharing time decreased. Studies showed that more than 50% of students occasionally felt pressure when dealing with their families during the pandemic, and most college students felt that their parents couldn’t understand them, that their communication with their parents was constrained, and even found that it was difficult to communicate with them [23]. During the pandemic period, family members would also have differences of opinions regarding the authenticity and seriousness of the pandemic news, whether to abide by the basic means of protection, and current affairs concerning the pandemic, which would cause conflicts, make college students unwilling to share with their families, and cause them to choose to isolate themselves.

Due to the close spatial distance, college students could communicate with their families when they encountered something, but they were not willing to share their difficulties or seek help and psychological comfort. Therefore, the close geographical location brought by the pandemic didn’t shorten the psychological distance, but rather made college students feel alienated from their families. During the pandemic period, college students were also faced with various academic pressures brought by the delay of the school term, such as the failure to carry out experimental operations and graduation as scheduled. In the short term, negative emotions were easy to accumulate, which might have led to changes in their mood and behavior [9]. Because family members couldn’t help solve many academic problems, college students would not choose to talk to their families or ask for advice. For personal problems, college students expressed they would be more likely to turn to a close friend for help [24].

When encountering difficulties, college students didn’t think family members supported each other, but they thought that they cared about each other. Due to the COVID-19 outbreak, college students communicated more with their families and paid more attention to the family, which was a positive outcome, but didn’t deeply influence the family dynamic. College students didn’t seek help or mutual support when they had problems, which might be related to college students’ poor mental health levels during the COVID-19 pandemic [25]. Although compared to older people who had mostly reported severe mental illness in the first few months of the COVID-19 outbreak [6], a survey on mental health problems of Chinese college students during the pandemic revealed that about 24.9% of the surveyed college students had obvious anxiety and depression, and 0.9% suffered from severe anxiety [26]. Different kinds of psychological stress, anxiety, and depression may affect the experience of family harmony.

As the COVID-19 pandemic is currently happening, the results of this study can be applied to interventions for family harmony and mental health among college students to reduce the negative impact of the COVID-19 pandemic.

The results of this study produced some guidelines for families. First, family members should understand each other. College students should undertake more housework, and their parents should try to understand their children’s psychological distress caused by the uncertainty of their time back to school [27]. Secondly, an effective communication mechanism should be established. All family members should consider others’ perspectives as much as possible, correct their mentality, and regulate their emotions. Everyone should seek common ground while reserving differences when treating different views and attitudes to reduce family conflicts. At the same time, interaction can be enhanced while staying at home by engaging in activities such as cooking, exercising, or other recreational hobbies. When parents get along with their children, they should respect each other to avoid high-pressure tension in the parent–child relationship. Thirdly, college students can try to communicate with their parents when they encounter difficulties and find ways to solve the problems together. Furthermore, empirical evidence suggests that family members, especially the younger ones, could considerably improve older people’s mental health status and, more generally, their subjective wellbeing [28]. Additionally, interactions with other family members helped older people to survive the early phase of the pandemic [29]. Therefore, college students could play a more critical family role via taking the responsibility of interacting with older family members and practicing the traditional virtue of filial piety. In doing so, college students may improve their mental health by the moral values attached to interactions with the older family members. Finally, when family conflicts cannot be solved, they should seek help from professional institutions in time.

There are also some suggestions for other interested parties. The public sector should use the influence of the media and the public to enhance the mutual trust between the public and the authorities, cut off the transmission path of rumors and other negative information, and create a positive public opinion environment [30]. The universities should release the teaching arrangements in a timely manner during the pandemic period to make sure that college students can understand the time back to school and the progress of the class, which avoids insecurity caused by information gaps and stabilizes the emotions of college students caused by academic problems [31]. Counselors and psychological monitors should pay active attention to the mental health status of college students during the pandemic period. At the same time, the official platform of the university should popularize the knowledge related to the pandemic, carry out mental health education during the pandemic situation, set up a psychological counseling service hotline or online consultation, and provide counseling and intervention for students with psychological distress to reduce their levels of depression, anxiety, and stress during the COVID-19 pandemic [1].

This study investigated changes in the experience of family harmony and family interaction from the perspective of college students, while other family members have different understandings and experiences [32]. Thus, future studies can investigate family harmony from the perspective of other family members to have a more comprehensive understanding of the changes in family interaction and their reasons. Although this study found the deterioration of family harmony and proposed relevant suggestions, there was no practical intervention. Therefore, future studies should design intervention programs according to the intervention design of this study and verify the effect of the program, allowing the study to have practical significance during the COVID-19 pandemic.

## 5. Conclusions

The college students’ experience of family harmony was significantly worse after the COVID-19 outbreak, especially on the dimensions of getting along, conflict, sharing, self-isolation, help-seeking, avoidance, and support-providing. Our study expands the study of family interaction and mental health during the COVID-19 pandemic. A variety of changes have been analyzed in this study and the related proposals were offered. Our study has also suggested some important future research areas for the COVID-19 pandemic.

## Figures and Tables

**Table 1 ijerph-19-06265-t001:** The college students’ experience of family harmony questionnaire (CSEFHQ).

Dimension	Item Number	Item	Cronbach’s α
Getting along	GE1	We don’t feel stressed at home.	0.91
	GE2	Family members don’t have to be careful when they communicate with each other.	
	GE3	Family members always get along with each other.	
	GE4	We seldom have family conflicts.	
	GE5	My family members love each other.	
	GE6	When there is a conflict in the family, the family members can be modest to each other.	
	GE7	Every member of the family is free to express his/her opinions.	
	GE8	There is always full of laughter among family members at home.	
	GE9	I feel that everyone in the family is backing each other.	
Conflict	CO1	Family members often quarrel with each other.	0.88
	CO2	Family members have a cold war with each other.	
	CO3	My family members are seldom gentle and considerate to each other.	
	CO4	Family members complain about each other when things go wrong.	
	CO5	Family members often blame and criticize each other.	
	CO6	I feel like I don’t want to stay at home.	
	CO7	The atmosphere at home is depressing and suffocating.	
Undertaking housework	UN1	We will discuss the division of housework.	0.88
	UN2	We take turns to share different housework in the family.	
	UN3	We do housework together.	
	UN4	We can share the housework together.	
	UN5	We all share family obligations.	
	UN6	We are willing to spend a lot of energy doing things at home.	
	UN7	Everyone in the family does his/her job.	
Sharing	SH1	We will try our best to spend time with our family members.	0.84
	SH2	I am very satisfied that my family spends time with me.	
	SH3	My family members take part in recreational activities together.	
	SH4	We share interesting stories together.	
	SH5	We will listen to each other’s opinions when we meet problems.	
	SH6	We will discuss and consult together when we encounter problems.	
	SH7	We’ll show each other our love.	
	SH8	We participate in things we are all interested in.	
Self-isolaion	SE1	We don’t have enough time to get along and communicate with each other.	0.88
	SE2	We prefer to do things separately rather than with the whole family.	
	SE3	We seldom consider the opinions of the rest of the family when we do things.	
	SE4	There is little time for family members to spend time with each other.	
	SE5	We don’t express our love for each other.	
Help-seeking	HE1	I can tell my family about my difficulties and troubles.	0.90
	HE2	I will discuss the solution with my family if I have a problem.	
	HE3	I can get comfort and help at home when I encounter difficulties.	
	HE4	I communicate with my family as soon as possible when something happens to me.	
	HE5	I will take the initiative to talk to my family.	
Avoidance	AV1	When I have something to worry about, I choose to take it alone.	0.87
	AV2	I never tell my family what’s on my mind.	
	AV3	I don’t talk to my family when I’m angry.	
	AV4	I don’t tell my family what happened.	
	AV5	When I come across something that makes me sad, it’s hard to talk to my family.	
	AV6	There’s no one to talk about my pain at home.	
Support-providing	SU1	I will care for my family members.	0.81
	SU2	I can help my family members when they are in trouble.	
	SU3	I will pay attention to my family members when they are in trouble.	
	SU4	I can give warmth and comfort to my family members when they need it.	
	SU5	I will support the ideas or decisions of other family members.	
	SU6	We can support each other in times of crisis.	
Indifference	IN1	My family and I don’t care about each other.	0.66
	IN2	The family members only care about themselves and ignore the family.	
	IN3	I’m self-centered and I don’t care about my family.	

**Table 2 ijerph-19-06265-t002:** Statistics of participants.

	Male	Female	Total	Rate
Freshmen	71	83	154	71.6%
Sophomores	20	30	50	23.3%
Juniors	5	6	11	5.1%
Total	96	119	215	
Rate	44.7%	55.3%		

**Table 3 ijerph-19-06265-t003:** The results of the paired sample *t*-test on the total score of CSEFHQ.

	M	SD	t	df	*p*
Pre-test	193.10	24.27	6.50	214	0.000
Post-test	185.51	24.96			

**Table 4 ijerph-19-06265-t004:** The results of paired sample *t*-tests on 9 dimensions of CSEFHQ.

Factor	Pre-Test Mean (SD)	Post-Test Mean (SD)	t	*p*
Getting along	31.76 (4.36)	30.57 (4.27)	5.116	0.000
Conflict	25.55 (3.08)	24.34 (3.42)	6.442	0.000
Undertaking housework	22.21 (4.00)	21.89 (3.79)	1.379	0.169
Sharing	27.81 (4.03)	26.60 (4.29)	5.414	0.000
Self-isolation	16.50 (2.80)	15.99 (2.79)	3.014	0.003
Help-seeking	16.93 (3.00)	16.03 (2.94)	5.353	0.000
Avoidance	19.31 (3.70)	18.11 (3.99)	6.010	0.000
Support-providing	21.73 (2.54)	20.82 (2.66)	5.818	0.000
Indifference	11.31 (1.27)	11.16 (1.22)	1.765	0.079

**Table 5 ijerph-19-06265-t005:** The results of paired sample *t*-tests on 56 items of the CSEFHQ.

Dimension	Item Number	Pre-Test Mean (SD)	Post-Test Mean (SD)	t	*p*
Getting along	GE1	3.61 (0.62)	3.50 (0.72)	2.008	0.046
	GE2	3.58 (0.70)	3.43 (0.80)	2.412	0.017
	GE3	3.53 (0.63)	3.39 (0.62)	3.124	0.002
	GE4	3.27 (0.82)	3.11 (0.78)	2.897	0.004
	GE5	3.64 (0.54)	3.54 (0.54)	2.641	0.009
	GE6	3.39 (0.68)	3.27 (0.68)	2.445	0.015
	GE7	3.59 (0.59)	3.46 (0.37)	3.326	0.001
	GE8	3.54 (0.61)	3.34 (0.64)	4.782	0.000
	GE9	3.62 (0.59)	3.52 (0.55)	2.856	0.005
Conflict	CO1	3.47 (0.72)	3.43 (0.71)	0.784	0.434
	CO2	3.74 (0.55)	3.64 (0.60)	2.556	0.011
	CO3	3.64 (0.61)	3.45 (0.65)	3.830	0.000
	CO4	3.57 (0.60)	3.34 (0.72)	4.623	0.000
	CO5	3.51 (0.64)	3.27 (0.70)	5.151	0.000
	CO6	3.75 (0.58)	3.51 (0.73)	5.343	0.000
	CO7	3.87 (0.37)	3.70 (0.54)	4.994	0.000
Undertaking housework	UN1	2.91 (0.82)	2.89 (0.80)	0.236	0.814
	UN2	2.97 (0.78)	2.95 (0.72)	0.324	0.746
	UN3	3.09 (0.81)	3.06 (0.74)	0.664	0.508
	UN4	3.17 (0.81)	3.18 (0.71)	−0.171	0.864
	UN5	3.27 (0.77)	3.24 (0.65)	0.695	0.488
	UN6	3.33 (0.69)	3.21 (0.68)	2.663	0.008
	UN7	3.47 (0.58)	3.37 (0.59)	2.428	0.016
Sharing	SH1	3.49 (0.70)	3.40 (0.61)	2.006	0.046
	SH2	3.63 (0.57)	3.47 (0.61)	4.234	0.000
	SH3	3.51 (0.67)	3.35 (0.71)	3.261	0.000
	SH4	3.58 (0.57)	3.36 (0.68)	0.249	0.804
	SH5	3.51 (0.56)	3.33 (0.62)	4.312	0.000
	SH6	3.47 (0.59)	3.40 (0.61)	2.062	0.040
	SH7	3.23 (0.80)	3.03 (0.81)	3.861	0.000
	SH8	3.39 (0.73)	3.26 (0.72)	2.651	0.009
Self-isolation	SE1	3.18 (0.93)	3.27 (0.83)	−1.237	0.217
	SE2	3.30 (0.74)	3.09 (0.82)	3.904	0.000
	SE3	3.63 (0.56)	3.45 (0.65)	4.154	0.000
	SE4	3.29 (0.76)	3.26 (0.71)	0.625	0.531
	SE5	3.10 (0.82)	2.92 (0.89)	2.854	0.005
Help-seeking	HE1	3.52 (0.70)	3.37 (0.68)	3.205	0.002
	HE2	3.45 (0.68)	3.20 (0.75)	4.546	0.000
	HE3	3.64 (0.60)	3.39 (0.67)	5.237	0.000
	HE4	3.07 (0.79)	3.06 (0.76)	0.249	0.804
	HE5	3.25 (0.80)	3.01 (0.85)	4.349	0.000
Avoidance	AV1	2.94 (0.86)	2.78 (0.93)	3.024	0.003
	AV2	3.21 (0.83)	2.89 (0.85)	5.066	0.000
	AV3	3.19 (0.81)	3.03 (0.81)	3.196	0.002
	AV4	3.34 (0.72)	3.10 (0.80)	4.437	0.000
	AV5	2.99 (0.92)	2.92 (0.88)	1.023	0.307
	AV6	3.65 (0.64)	3.39 (0.73)	5.341	0.000
Support-providing	SU1	3.68 (0.54)	3.58 (0.52)	2.667	0.008
	SU2	3.53 (0.59)	3.41 (0.60)	2.878	0.004
	SU3	3.63 (0.50)	3.47 (0.60)	4.176	0.000
	SU4	3.61 (0.57)	3.47 (0.57)	3.783	0.000
	SU5	3.55 (0.53)	3.40 (0.56)	3.937	0.000
	SU6	3.76 (0.49)	3.52 (0.52)	5.234	0.000
Indifference	IN1	3.81 (0.48)	3.77 (0.48)	1.100	0.273
	IN2	3.76 (0.55)	3.77 (0.49)	−0.112	0.911
	IN3	3.74 (0.52)	3.62 (0.57)	3.219	0.001

## Data Availability

The data presented in this study are available upon request from the corresponding author. The data are not publicly available, as they are a part of a developing dataset that will be used in the future for different studies.

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
