# Peer review of "Analysis of the Difference in College Students’ Experience of Family Harmony before and after the COVID-19 Outbreak"

_ijerph, 2022, doi:10.3390/ijerph19106265_

Round 1

Reviewer 1 Report

Thank you for submitting the manuscript to this journal. You have chosen an interesting subject to do this study. Some issues need to be clarified. I have summarized these concerns below: the following comments need to be taken with care in order to improve the quality of the manuscript for publishing.

ABSTRACT

The conclusions presented in the summary do not respond to the stated objective. Authors need to redraft conclusions.

KEYWORDS

Keywords must be in alphabetical order.

The COVID-19 outbreak eliminate “the”.

Review the keywords, and try to make them MeSH.

INTRODUCTION

The inserted reference must be separated from the word.

The authors only use eight references in the introduction. It is important that the authors carry out a more in-depth review of the literature.

It would be interesting for the authors to include a paragraph with the recent research carried out on the subject, highlighting what this article can contribute with respect to what is already known.

The objective must be written in a clear and direct way, please, review the wording of the objective.

METHODS

Indicate the study design, include this section in the method.

Indicate the calculation of the sample size.

What were the inclusion and exclusion criteria of the participants?

What was the guide used based on the design to guarantee quality.

They need to provide more information on data collection. Include the data collection section

The authors must include a subsection on ethical aspects and data analysis.

Table 1 should be included in annexes, not in the main text of the manuscript.

RESULTS

Include in the results section, the sociodemographic characteristics of the sample and eliminate table 2.

3.2. The difference and analysis on 9 dimensions of CSEFHQ before and after the COVID-19 outbreak. Eliminate 9.

DISCUSSION

Avoid the first person plural in the wording.

The authors do not include the limitations of the study.

It would be necessary to include new future lines to investigate, after the completion of the study.

It would be convenient to indicate the implications of the manuscripts.

CONCLUSION

The conclusions must be clear, direct and respond to the stated objective, for this reason the authors must review them.

REFERENCES

References must be adapted to the format of the journal.

Author Response

As your kind suggestions, all the comments have been adopted and modifications have been conducted in the revised manuscript. The comments are replied specificly in the annexes. Thanks again for your constructive suggestions.

Reviewer 2 Report

Thanks for inviting me to review this manuscript. This paper investigates university students’ experience of family harmony during the early phase of the pandemic. I must congratulate the authors because even for an extremely fussy reviewer like me, this paper seems interesting. Even so, I believe the paper could be improved in the following ways:

Major comments:

  1. An “experience” is very much an objective issue. Maybe a “perception of family harmony” or “perceived family harmony” would serve the purpose of this paper better. I am not sure whether this is the definition of family harmony “Family harmony is an ideal state of family relations, which refers to the harmonious coexistence in family life”. If so, the author should clarify the conceptualisation of this notion, i.e., how it can be measured. The authors introduced the five dimensions in the questionnaire; however, no explanation is provided. If this is not the definition, please define it. Also, in lines 63-67, the author mentioned quite a few other notions. What makes the notion of family harmony distinct from the others? Does it have a special focus? Does it describe a special kind of relationship between family members? These need to be clarified.
  2. Line 72-74, the author stated that “There are no longitudinal studies comparing changes at individual mental health levels before and after the COVID-19 outbreak”. This is not true. Please see Liu et al. (2022), which uses three-wave panel data.
  3. Line 93, there is no point to see the alpha value for the whole scale, please provide the Cronbach’s alpha for each of the constructs in Table 1.
  4. It would be much better to conduct an EFA (use PAF with oblimin rotation). To check if all the items correspond with their hypothesised constructs. It could be very interesting to see if there are more factors than expected.
  5. Table 5. It is pointless to show the t-test on each item because only constructs are meaningful in a likert scale. Standing alone, they just don’t make any sense. So, please rewrite section 3.3 thoroughly.
  6. It may be interesting to put your study in a wider context, about how family relationships may have changed during the pandemic. For example, Gadassi Polack et al., (2021) revealed that the large-scale quarantine has changed the types of interaction between adolescents and others. Lee et al. (2020) investigated how the pandemic and its containment interventions have changed family life in Korea. Liu et al. (2021) found that interactions with other family members helped older people to survive the early phase of the pandemic.

Minor comments:

  1. Please add a reference for this “Experts predict that the COVID-19 pandemic won’t come under control until 2025” (page 2, line 77)

Reference

Gadassi Polack, R., Sened, H., Aubé, S., Zhang, A., Joormann, J., & Kober, H. (2021). Connections during crisis: Adolescents’ social dynamics and mental health during COVID-19. Developmental Psychology, 57(10), 1633.

Lee, J., Chin, M., & Sung, M. (2020). How has COVID-19 changed family life and well-being in Korea?. Journal of Comparative Family Studies, 51(3-4), 301-313.

Liu, Q., Liu, Y., Zhang, C., An, Z., & Zhao, P. (2021). Elderly mobility during the COVID-19 pandemic: A qualitative exploration in Kunming, China. Journal of Transport Geography, 96, 103176.

Liu, Q., Liu, Z., Lin, S., & Zhao, P. (2022). Perceived accessibility and mental health consequences of COVID-19 containment policies. Journal of Transport & Health, 101354.

Author Response

As your kind suggestions, all comments have been carefully considered and modifications have been conducted in the revised manuscript. We have revised some statements in the revised manuscript. The comments are replied specificly in the annexes. Thanks again for your constructive suggestions.

Round 2

Reviewer 1 Report

Congratulations the manuscript has been improved considerably.

Author Response

Thanks very much for your suggestions.

Reviewer 2 Report

Comment 1

Empirical evidence suggests that family members, especially the younger ones could considerably improve older people's mental health status and, more generally, their subjective wellbeing (see Girdhar et al., 2020; Settersten et al., 2020). In the discussion section, (maybe line 280) you could add a sentence about this. Considering this, "Sencondly, college students could play a more critical family role via taking the responsibility of interacting with older family members and practicing the traditional virtue of filial piety. In doing so, college students may improve their mental health by the moral values attached to interactions with the older ones".

Reference

Girdhar, R., Srivastava, V., & Sethi, S. (2020). Managing mental health issues among elderly during COVID-19 pandemic. Journal of geriatric care and research, 7(1), 32-35.
Settersten Jr, R. A., Bernardi, L., Härkönen, J., Antonucci, T. C., Dykstra, P. A., Heckhausen, J., ... & Thomson, E. (2020). Understanding the effects of Covid-19 through a life course lens. Advances in Life Course Research, 45, 100360.

Comment 2

You could place your research into a broader context, comparing college students with other population groups. For example, in Line 269, you could try, "Although compared to older people who had mostly reported severe mental illness in the first few months of the Covid-19 outbreak (Liu et al., 2022), a survey on mental health problems of Chinese college students during the pandemic revealed that ..."

Reference

Liu, Q., Liu, Z., Lin, S., & Zhao, P. (2022). Perceived accessibility and mental health consequences of COVID-19 containment policies. Journal of Transport & Health, 101354.

Author Response

As your kind suggestions, all the comments have been adopted and modifications have been conducted in the revised manuscript. The comments are replied specificly in the annexes. Thanks again for your constructive suggestions.

This manuscript is a resubmission of an earlier submission. The following is a list of the peer review reports and author responses from that submission.